# Membrane Lipids and Osmolytes in the Response of the Acidophilic Basidiomycete *Phlebiopsis gigantea* to Heat, Cold, and Osmotic Shocks

**DOI:** 10.3390/ijms25063380

**Published:** 2024-03-16

**Authors:** Elena A. Ianutsevich, Olga A. Danilova, Olga A. Grum-Grzhimaylo, Vera M. Tereshina

**Affiliations:** 1Winogradsky Institute of Microbiology, Research Center of Biotechnology of the Russian Academy of Sciences, 33, bld. 2 Leninsky Ave., 119071 Moscow, Russia; noitcelfer@mail.ru (O.A.D.); v.m.tereshina@inbox.ru (V.M.T.); 2White Sea Biological Station, Faculty of Biology, Lomonosov Moscow State University, 1–12 Leninskie Gory, 119234 Moscow, Russia; olgrgr@wsbs-msu.ru; 3Laboratory of Genetics, Plant Sciences Group, Wageningen University, Droevendaalsesteeg 1, 6708PB Wageningen, The Netherlands

**Keywords:** acidophilia, heat shock, cold shock, osmotic shock, trehalose, phosphatidic acids

## Abstract

Previously, we found for the first time the participation of osmolytes in adaptation to acidic conditions in three acidophilic fungi. Because trehalose can protect membranes, we hypothesized a relationship between osmolyte and membrane systems in adaptation to stressors. In the mycelium of *Phlebiopsis gigantea*, the level of osmolytes reaches 8% of the dry mass, while trehalose and arabitol make up 60% and 33% of the sum, respectively. Cold shock does not change the composition of osmolytes, heat shock causes a twofold increase in the trehalose level, and osmotic shock leads to a marked increase in the amount of trehalose and arabitol. Predominance of phospholipids (89% of the sum) and low proportions of sterols and sphingolipids are characteristic features of the membrane lipids’ composition. Phosphatidic acids, along with phosphatidylethanolamines and phosphatidylcholines, are the main membrane lipids. The composition of the membrane lipids remains constant under all shocks. The predominance of linoleic (75% of the sum) and palmitic (20%) acids in phospholipids results in a high degree of unsaturation (1.5). Minor fluctuations in the fatty acid composition are observed under all shocks. The results demonstrate that maintaining or increasing the trehalose level provides stability in the membrane lipid composition during adaptation.

## 1. Introduction

Extremophilic fungi play a key role in the functioning of extreme ecosystems because of their highly flexible lifestyles and stunning ecological and morphological versatility [1]. The study of extremophiles is necessary for understanding the biochemical foundations of life on Earth and the properties of biomolecules that allow them to master extreme habitats [2]. Such studies are also of interest for astrobiology, since fungi are the key organisms in the search for the limits of life on Earth and in the Universe and ecology—for the preservation of biocenoses despite abrupt climate changes and increased anthropogenic influence. Extremophiles are used for bioremediation and biofuel production and are also sources of new biologically active compounds and extremozymes [3,4,5].

Acidophilic fungi are a group of extremophiles capable of carrying out their development cycle under acidic environmental conditions, whereas most fungi prefer near-neutral pH values. Acidophiles include fungi that have an optimum growth at pH below 4.0, but are not capable of growth under neutral conditions, whereas acidotolerants grow under acidic, neutral, or even slightly alkaline conditions [1]. Natural habitats with acidic conditions (pH 3.0–4.0) include acidic soils, lakes, swamps, and peatlands [6]. Extremely acidic pH values < 3.0 are typical for terrestrial hydrothermal vents, volcanic lakes, acidic coal mine dumps, mine drainage, and industrial wastewater.

The microbial community of such habitats is represented by prokaryotes (archaea and bacteria) and eukaryotes, which together form biofilms under natural conditions [7,8]. The main representatives of eukaryotes in acidic habitats are algae (diatomaceous, green and red), protozoa, and fungi [9,10,11].

Acidic environmental conditions affect the membrane charge, digestibility of substrates, function of proteins, and toxicity of metal ions [12]. One of the main features of adaptation to acidic environments is maintaining a neutral intracellular pH through the use of proton pumps that provide the efflux of hydrogen ions out of the cell [13]. In fungi, the intracellular pH regulation system includes vacuolar-type ATPases (V-ATPase) and a P-type proton pump Pma1, which act in conjunction with a large number of other transporters [14]. For example, the acidophilic fungus *Acidiella bohemica* maintains intracellular pH homeostasis by creating a positive potential on the cytoplasmic membrane, removing protons from the cell using antiporters and cotransporters, creating a buffer in the cytoplasm using glutamic and phosphoric acids, and degrading organic acids [15]. The enzymes of acidophiles have certain modifications that contribute to their stability under acidic conditions. Thus, in comparison with neutrophilic and alkaliphilic fungi, the crystal structure of endo-1,4-b-xylanase (optimum at pH 3.2) from the acidophile *Scytalidium acidophilum* contains significantly fewer salt bridges and hydrogen bonds, has a negative surface charge, and has a specific conservative sequence of amino acids in the active site [16]. Using the example of the neutrophilic fungus *Ustilago maydis*, it was found that adaptation to acidic, alkaline environmental conditions and osmotic stress is regulated by the transcription factor NRG1 [17].

Numerous studies of the fungal response to a variety of abiotic factors have shown that a key element in the adaptation to stress is the protection of cell membranes and macromolecules by changing the composition of membrane lipids and cytoplasmic osmolytes and the synthesis of chaperones and antioxidants [18,19,20]. Osmolytes include low-molecular-weight organic compounds of various chemical natures, which are used to protect cells against stress effects [21,22]. It is important to note that osmolytes are not only compatible solutes [23], but also have neutralizing and cytoprotective abilities, protecting both macromolecules and cell membranes. On the basis of these data, we assume that changes in the osmolyte and membrane systems of the cell can be interconnected and need to be studied in parallel. We previously suggested that osmolytes and membrane lipids may be involved in adaptation to acidic conditions by protecting the cytoplasmic membrane from an aggressive external environment. Our research confirmed this assumption: for the first time, a large amount of trehalose was found in the mycelium of three acidophilic fungi—*Sistotrema brinkmannii*, *Phlebiopsis gigantea*, and *Mollisia* sp.—under optimal conditions (pH 4.0) [24,25].

Acidophiles, like all other fungi, are exposed to additional stress factors (heat, cold, osmotic) under natural conditions. Previous data on the effects of trehalose protection [24,25] suggest that acidophiles can use the cell membrane and osmolytic systems to adapt to a variety of abiotic stressors.

The purpose of this research is to study the composition of osmolytes and membrane lipids in the acidophilic fungus *P. gigantea* under heat, cold, and osmotic shocks.

## 2. Results

None of the studied shock factors led to any noticeable morphological changes in the fungal mycelium. Note that the protocol of experiments with heat shock (HS) and cold shock (CS) differed from those with osmotic shock (OS). During HS and CS, Petri dishes were transferred to the appropriate temperature conditions, while Control 1 continued to grow under optimal conditions. In the case of OS, mycelium on a cellophane substrate was transferred to a medium containing 0.25 M sodium chloride, and the Control 2 mycelium was also transferred to a fresh medium. The fungus grew best on a salt-free medium (growth rate of 3 mm/day), whereas in the presence of 0.125 M NaCl, the growth rate dropped by 3.5 times, and an increase in salt concentration to 0.25 M led to a stop in the growth process, so this concentration was used to create OS.

### 2.1. Composition of the Cytosol Carbohydrates and Polyols in the Mycelium of the Fungus under the Action of Heat, Cold, and Osmotic Shocks

Under optimal conditions, carbohydrates and polyols (CaP) of *P. gigantea* mainly consisted of trehalose, glucose, and arabitol carbohydrates (Control 1) (Figure 1a), with trace amouts of glycerol, erythritol, inositol, and mannitol (less than 0.05% of the dry mass each, not shown). The amount of CaP reached 8% of the dry mass, with trehalose being predominant (61% of the sum), and the proportions of arabitol and glucose were 34% and 5%, respectively (Figure 1b). CS did not cause noticeable changes in either the composition or the proportions of CaP, whereas HS resulted in a twofold increase in the amount of trehalose, reaching 78% of the sum, while the proportion of arabitol decreased by half. At the same time, the amount and proportions of glucose did not change during both HS and CS.

Compared with Control 1, in Control 2, which was transferred to fresh medium, the amount of CaP was halved, and their ratios changed due to a decrease in the amount of trehalose (Figure 1a). The proportion of arabitol increased from 33% to 47%, while the proportion of trehalose decreased from 60% to 38% (Figure 1b). OS led to an increase in the amount of CaP by more than one and a half times due to an increase in the level of arabitol to 4% and trehalose to 3.4% of the dry mass, while their ratio, compared to Control 2, changed little: the proportion of glucose decreased from 10% to 5% of the sum.

### 2.2. Composition of Membrane and Storage Lipids in the Mycelium of the Fungus under the Action of Heat, Cold, and Osmotic Shocks

The amount of membrane and storage lipids under optimal conditions (Control 1) and under the action of HS and CS was approximately 4% of the dry mass (Table 1). Transfer to fresh medium (Control 2) led to an increase in the amount of storage lipids compared with Control 1, while OS did not change their amount.

*P. gigantea* membrane lipids under optimal conditions (Control 1) were phospholipids (89% of the sum), sterols (7%), and sphingolipids (SL) (4%). The main membrane lipids of the fungus under optimal conditions were phosphatidylethanolamines (PE) (31% of the sum), phosphatidylcholines (PC) (27%), and phosphatidic acids (PA) (17%) (Figure 2a). The proportions of cardiolipins (CL), phosphatidylserines (PS), phosphatidylinositols (PI), and lysophosphatidylethanolamines (LPE) did not exceed 5%. Traces of lysophosphatidylcholines (LPC) and an unidentified lipid (X) were also found. Under the influence of HS and CS, no noticeable changes in the composition of membrane lipids were observed. HS led to a slight decrease in the amount of PE but not in its proportion. In contrast, an increase in the proportion of sterols can be observed under HS and CS, but their amount remained constant (Figure 2b). In Control 2, compared with Control 1, noticeable changes were found in both the amount and proportions of membrane lipids due to an increase in the amount (twofold) and proportion (from 17 to 27%) of PA. However, compared with Control 2, OS resulted in a slight increase in the proportion of PC (about 15%) against the decrease in the proportion of PC, but the amount of these phospholipids did not change significantly.

The storage lipids of *P. gigantea* in Control 1 are represented mainly by MAG, DAG, and TAG in all variants of the experiment (Figure 3a), with MAG and DAG being predominant (30% of the sum). The minor components included FFA, sterol esters, and unidentified lipid Y. No significant changes were observed under the influence of CS and OS. The effect of HS was accompanied by a slight increase in the proportion of TAG (Figure 3b). The levels of MAG and DAG increased in Control 2, compared with Control 1. However, under OS, neither the amount nor the ratio of storage lipids changed, compared with Control 2.

### 2.3. Fatty Acid Composition of Phospholipids in the Mycelium of the Fungus under Heat, Cold, and Osmotic Shocks

The fatty acid composition of *P. gigantea* phospholipids in all variants of the experiment was dominated by linoleic acid (C18:2), the proportion of which varied between 70–75% of the sum, and palmitic acid (C16:0) (18–20% of the sum). The remaining fatty acids were present in trace amounts (Figure 4a). Under the influence of all types of shocks, a relative permanence of the fatty acid composition of the phospholipid fraction is observed, which causes a constant degree of unsaturation (DU) of approximately 1.5 (Figure 4b). A slight increase in the proportion of linoleic acid and a simultaneous decrease in the proportion of myristoleic acid after OS did not change DU. An increase in the proportion of palmitic acid can be observed against the background of a decrease in linoleic acid at HS, which leads to a slight decrease in DU (by 4%).

## 3. Discussion

Extremophilic fungi have acquired resistance to at least one unfavorable environmental factor. An important component of such resistance is the protection of cell membranes and macromolecules, which can be performed both by changing the composition of membrane lipids and by using protective compounds (osmolytes), which are represented in fungi by trehalose and polyols [26]. All osmolytes are multifunctional compounds. The disaccharide trehalose has protective, reserve, antioxidant, regulatory, and chaperone functions [27,28,29,30,31]. Polyols have osmoprotective and antioxidant properties, regulating the redox balance of the cell [26,32,33,34].

Resistance to other types of stressors can occur in non-extremophilic fungi under the action of one type of stress [35]. The phenomenon of cross-resistance is based on the general environmental stress response or core environmental response, which is regulated by transcription factors Msn2p and Msn4p, which bind to a DNA site called the Stress Response Element (STRE) [30,35,36,37,38]. The phenomenon of cross-protection suggests that different stressors require universal responses: adjustment of energy metabolism, production of protective proteins (such as HSP) or small protective molecules (osmolytes, such as glycerol or trehalose). Since trehalose is present in the mycelium of the acidophiles in a significant amount, it can be assumed that the fungal cells will be protected from many types of stressful effects. In this research, it was shown that in the control variant, approximately 5% of trehalose and 2.5% of arabitol were present in *P. gigantea* cells (Figure 1). In Control 2, which was transferred to fresh medium, the amount of CaP was halved, compared with Control 1, and their ratios changed due to a decrease in the amount of trehalose. We assume that, being a reserve carbohydrate, trehalose is used up due to the activation of growth processes on fresh media. In response to CS (6 h at 5 °C), the composition of CaP does not change, whereas HS leads to an almost twofold increase in the amount of trehalose, and OS leads to a noticeable increase in the amount of arabitol and trehalose. These data indicate the general purpose of trehalose as a protector under all types of stress and the need for polyols to adapt to OS. It should be noted that the common characteristic features of the response of extremophiles and non-extremophilic fungi to stressors are a rapid increase in the level of trehalose during HS and polyols during OS [39]. From the standpoint of cross-resistance, adaptation to acidic conditions can provide protection against other types of stress. Interestingly, despite the high content of trehalose in the mycelium, a sharp increase in its level occurs during HS, i.e., an additive response is observed. The same data were obtained by us on the example of the alkaliphilic fungus *Sodiomyces tronii*, in which, under optimal conditions, approximately 10% of trehalose was observed in the mycelium, and with HS, its amount increased to 16% [40]. It can be assumed that trehalose protects different cell components depending on the type of stress. Probably, in alkaliphiles and acidophiles, this is mainly a cytoplasmic membrane in contact with an aggressive external environment that needs protection, and during HS, CS, and OS, these are macromolecules and all cell membranes, which are additionally subjected to dehydration and oxidative effects [18].

In archaea, unlike bacteria and fungi, the impermeability of membranes to protons is associated with the presence of specific tetraester lipids with simple ether bonds that are resistant to acid hydrolysis [41]. However, no specific membrane lipids were found in the three acidophilic fungi studied. A characteristic feature of *P. gigantea* membrane lipids is a low proportion of sterols and SL and a predominance of phospholipids (89% of the sum), as well as a high proportion of non-bilayer phospholipids PE and PA (Figure 2). Interestingly, the composition of membrane lipids during CS, HS, and OS does not change significantly. In addition, we found no noticeable changes in the composition of the fatty acids of the *P. gigantea* phospholipid fraction under all types of stress factors (Figure 4). This fungus demonstrates an unusual fatty acid composition, where polyunsaturated linoleic acid (75% of the sum) and saturated palmitic acid (20%) are predominant, and the remaining fatty acids are minor components. Thus, a high degree of unsaturation of the phospholipid fraction can be observed (about 1.5), which is usually characteristic of psychrophilic fungi [42,43]. It has also been previously shown that in *P. gigantea*, the degree of unsaturation remained constant both in growth dynamics and under unfavorable pH conditions [25]. The data obtained suggest that the mechanism for changing the degree of unsaturation is not involved in the adaptation of *P. gigantea* to either the pH factor or HS, CS, or OS. In contrast, in non-extremophilic fungi, a common pattern in response to CS is an increase in DU [42,43].

In our studies, we have shown that PAs, which are minor components in non-extremophiles [44], are present in extremophiles, either as predominant or as one of the main membrane lipids [24,25,45,46,47]. PAs are multifunctional and poorly studied compounds, being a key metabolite of lipogenesis and performing regulatory and structural functions [48,49]. Phospholipase D, localized in membranes, is involved in phospholipid signaling by controlling the level of PAs, which are secondary messengers of many kinases, allowing control of processes such as cytoskeletal rearrangement, reproduction and survival, vesicular intracellular transport, and endo- and exocytosis [50]. It is known that, depending on the conditions, PAs can exhibit both bilayer and non-bilayer lipid properties [50]. Thus, under slightly acidic conditions in the presence of divalent ions, for example, in the Golgi apparatus, they form type II micelles, but at a neutral pH in the absence of ions, PAs exhibit the properties of bilayer lipids. It is assumed that PAs can control transport from the Golgi apparatus and EPR, endocytosis, and exocytosis due to the ability to aggregate into microdomains, which leads to bending of membranes, resulting in vesicle formation. Previously, we have shown for the first time a high proportion of PA in various extremophilic fungi: thermophiles [45], alkaliphiles [46], acidophiles [24,25], and xerophiles [47]. Acidophiles, unlike neutrophils, are also characterized by the presence of PA as one of the main phospholipids, along with PE, PC, and sterols, which also confirms their role in extremophilia [24,25]. These data suggest that extremophile membranes have a special structure compared with non-extremophile membranes. They are characterized by a high proportion of non-bilayer lipids—PA and PE. In the studied fungus, the proportion of these phospholipids was 48% of the sum membrane lipids (Figure 2).

To study the relationship between changes in the osmolyte and membrane systems of the cell, we analyzed the composition of lipids and osmolytes under the action of HS, CS, and OS. It was found that, under the influence of CS, HS, and OS, if the level of trehalose did not fall or rise (Figure 1), the composition of membrane lipids (Figure 2) and their fatty acids (Figure 4) remained stable. Previously, we demonstrated that prolonged cultivation of *P. gigantea* under unfavorable pH conditions of 2.6 and 5.0 led to a noticeable decrease in the amount of trehalose and alterations in the composition of membrane lipids, whereas in *Mollisia* sp., the amount of osmolytes either did not decrease (at pH 6.0) or increased (at pH 2.6), and changes in the composition of membrane lipids were insignificant [25]. The only instance when the level of trehalose decreased significantly in this study was the difference between Control 1 and Control 2. In this case, a significant change in the composition of membrane lipids was observed, the proportion of PA increased, while the proportion of PE decreased, against the background of a decrease in the amount of trehalose, indicating the interrelation between changes in the membrane and osmolyte systems of the cell. Data supporting this assumption were also obtained in a comparative study of changes in the composition of membrane lipids in HS in trehalose of deficient and wild yeast strains of *Schizosaccharomyces pombe* [20]. During HS, trehalose-deficient yeast undergo much more extensive rearrangements in the composition of membrane lipids than wild-type strains that accumulate large amounts of trehalose.

Thus, the results obtained suggest that maintaining or increasing the level of trehalose can ensure the stability of the composition of membrane lipids under various stress impacts. Cross-resistance based on trehalose protection and consistency of membrane lipid composition indicates polyextremophilicity. The obtained data indicate the interconnection between changes in the cell osmolytic and membrane systems in the acidophilic fungus *P. gigantea*.

## 4. Materials and Methods

### 4.1. Objects of Study and Cultivation Protocol

Basidiomycete fungus *Phlebiopsis gigantea* (Fr.) Jülich (Phanerochaetaceae, Polyporales, Incertae sedis, Agaricomycetes, Agaricomycotina, Basidiomycota, Fungi) was isolated from peat from a depth of 0.5 m of the transitional aapa-type peatland, identified by ITS rDNA region, and deposited at GenBank with the accession number JQ780612 [51], and it was also deposited in the All-Russian Collection of Microorganisms (VKM) under number VKM F-4932.

The fungi were cultivated on a standard agarized medium based on malt extract, 17 g/L (“Condalab”), with citrate phosphate buffer at optimal pH 4.0 (pH tolerance range 2.6–6.0) at an optimum temperature of 24–25 °C during 10–14 days and stored at 5–8 °C. Agarized media consisted of two components: (1) the citrate-phosphate buffer component with 4.0 pH value and the nutrient component, containing 34 g/L of the malt extract (Condalab) with 40 g/L of agar. Components (1) and (2) were autoclaved separately at 121 °C for 20 min, cooled down to 55 °C, and then mixed in 1:1 ratio under sterile conditions, resulting in the final concentrations of the complete medium: 17 g/L malt extract, 20 g/L agar. The inoculum was grown on agarized medium on Petri dishes for 4 days (pH 4.0, 24–25 °C). For the plates’ inoculation, 1 × 1 mm fragments of mycelium were used, cut from the actively growing edge of the colony.

The dependence of the growth rate on the concentration of NaCl in the medium (0, 0.125, 0.25, 0.375, 0.5 M) was determined at the optimal pH and temperature (pH 4.0, 24–25 °C). The diameter of the colonies was measured in two perpendicular directions every 3–4 days until the colony reached the edges of the plate in one of the Petri dishes. The growth rate was determined as mm/day during the period of linear growth.

For biochemical studies of lipids and osmolytes, *P. gigantea* was pre-grown as follows. The mycelium agar plugs were cut from the parental agar slant culture and used to inoculate several 90 mm Petri dishes with the pH 4.0 medium, which was further cultivated at 24–25 °C for 7–10 days. The mycelium fragments were then cut from the actively growing edge of the colony and used to inoculate the appropriate number of 90 mm Petri dishes with a cellophane-coated medium. *P. gigantea* was cultivated for 14 days under optimal conditions (pH 4.0, 24–25 °C).

For HS, Petri dishes were transferred to 37.5 °C and incubated for 3 h; for CS, they were transferred to 5 °C for 6 h. Corresponding control variant (Control 1) was grown for the same time under optimal conditions. To create an OS, mycelium on a cellophane disk was transferred to Petri dishes with fresh media (pH 4.0) containing 0.25 M NaCl and incubated for 6 h under optimal temperature 24–25 °C. Corresponding control variant (Control 2) was transferred to Petri dishes with fresh media (pH 4.0) without NaCl and incubated for 6 h under optimal temperature 24–25 °C. The mycelium was removed from the cellophane support using a scalpel, weighed, and stored at −21 °C. Dry biomass was determined gravimetrically.

### 4.2. Lipids, Sugars and Polyols Analysis

Lipids, carbohydrates, and polyols analyses were performed as described earlier [24]. Briefly, lipids were extracted by the Nichols method [52] with phospholipase-deactivating isopropanol, separated using two-dimensional (polar lipids) or one-dimensional (neutral lipids) thin-layer chromatography (TLC) [53] and quantified using standard compounds by densitometry method (DENS software, version 5.1.0.2). To study the composition of fatty acids, the polar lipid fraction was isolated using one-dimensional TLC. The polar lipid spots at the start were scraped out and eluted with a mixture of chloroform: methanol (1:1), then the extract was evaporated and methanolysis was carried out using 2.5% H_2_SO_4_ in methanol for 2 h at 70 °C. The obtained methyl esters were analyzed by gas-liquid chromatography (GLC) on a Kristall 5000.1 gas-liquid chromatograph (Chromatec, Yoshkar-Ola, Russia) with an Optima-240, 60 m × 0.25 µm × 0.25 mm capillary column (Macherey-Nagel GmbH&Co, Düren, Germany). The temperature program used was from 130 to 240 °C at a rate of 5–6 °C/min. Identification was carried out using the Supelco 37 Component FAME Mix mixture of individual fatty acid methyl esters (Supelco, Bellefonte, PA, USA). The degree of unsaturation of the phospholipids (DU) was calculated according to the following equation [44]:DU = 1.0 × (% monoene FA)/100 + 2.0 × (% diene FA)/100 + 3.0 × (% triene FA)/100

To determine the soluble carbohydrate composition of the mycelium, sugars were extracted with boiling water for 20 min four times. Proteins were removed from the resulting total extract [54]. The carbohydrate extract was further purified from charged compounds using a combined column with the Dowex-1 (acetate form) and Dowex 50 W (H^+^) ion exchange resins. Carbohydrate composition was determined by GLC using trimethylsilyl sugar derivatives obtained from the lyophilized extract [55]. The internal standard was α-methyl-D-mannoside (Merck). Chromatography was carried out on a Kristall 5000.1 gas chromatograph (Chromatec, Yoshkar-Ola, Russia) with a ZB-5 30 m, 0.32 mm, 0.25 μm capillary column (Phenomenex, Torrance, CA, USA). The temperature was increased from 130 to 270 °C at a rate of 5–6 °C/min. Glucose, mannitol, arabitol, inositol, and trehalose (Sigma, St. Louis, MO, United States) were used as standards.

### 4.3. Statistical Analysis

The experiments were carried out in triplicate, *n* = 3. The post hoc Dunnett test was used for pairwise comparison between Control 1 and HS or CS, Control 2 and OS, Control 1 and Control 2. On all graphs, mean values ± SEM (standard error of the mean) are plotted. Statistically significant difference (*p* ≤ 0.05) is indicated by (*)—between Control 1 and HS or CS; by (#) between Control 2 and OS; by (†) between Control 1 and Control 2.

## Figures and Tables

**Figure 1 ijms-25-03380-f001:**
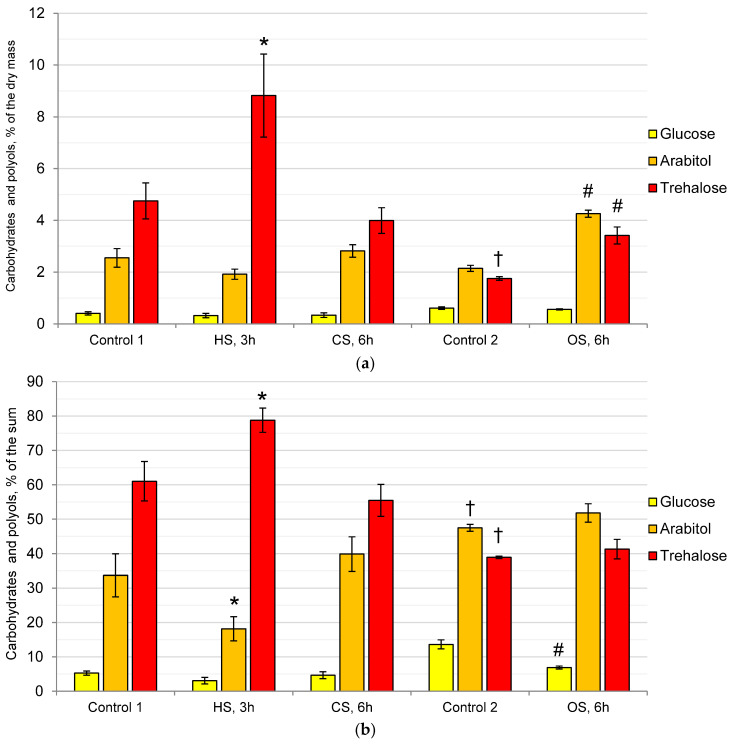
The main carbohydrates and polyols of *P. gigantea*: (**a**)—% of the dry mass; (**b**)—% of the sum. Control 1 and Control 2—optimal conditions, HS—heat shock, CS—cold shock, OS—osmotic shock. Means ± SEM are displayed, *n* = 3, SEM—standard error of the mean. Statistically significant difference (*p* ≤ 0.05) is indicated by (*)—between Control 1 and HS or CS; by (#) between Control 2 and OS; by (†) between Control 1 and Control 2.

**Figure 2 ijms-25-03380-f002:**
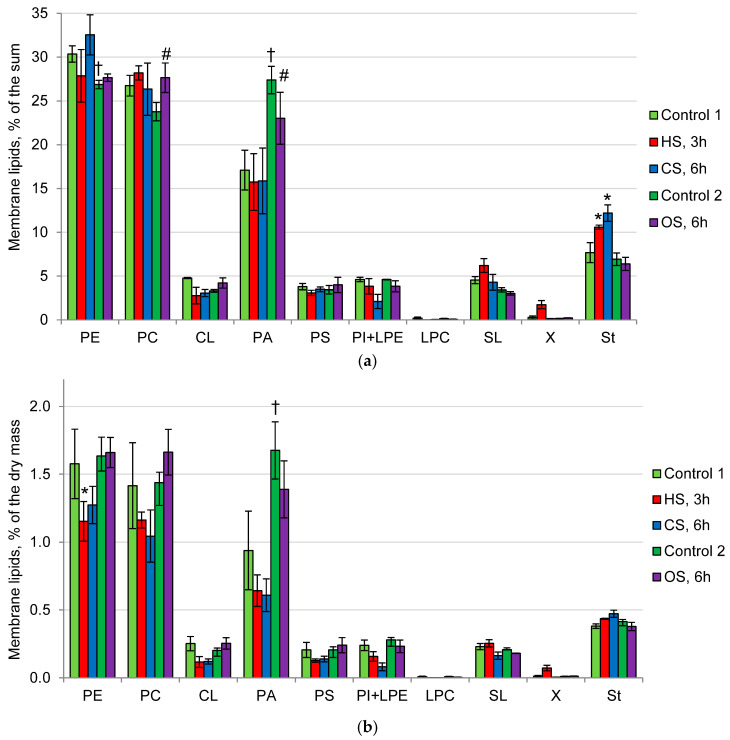
The profile of *P. gigantea* membrane lipids: (**a**)—% of the dry mass; (**b**)—% of the sum. Means ± SEM are displayed, *n* = 3, SEM—standard error of the mean. Statistically significant difference (*p* ≤ 0.05) is indicated by (*)—between Control 1 and HS or CS; by (#) between Control 2 and OS; by (†) between Control 1 and Control 2. PE—phosphatidylethanolamines, PC—phosphatidylcholines, CL—cardiolipins, PA—phosphatidic acids, PS—phosphatidylserines, PI—phosphatidylinositols, LPE—lysophosphatidylethanolamines, LPC—lysophosphatidylcholines, SL—sphingolipids, X—unidentified lipid, St—sterols.

**Figure 3 ijms-25-03380-f003:**
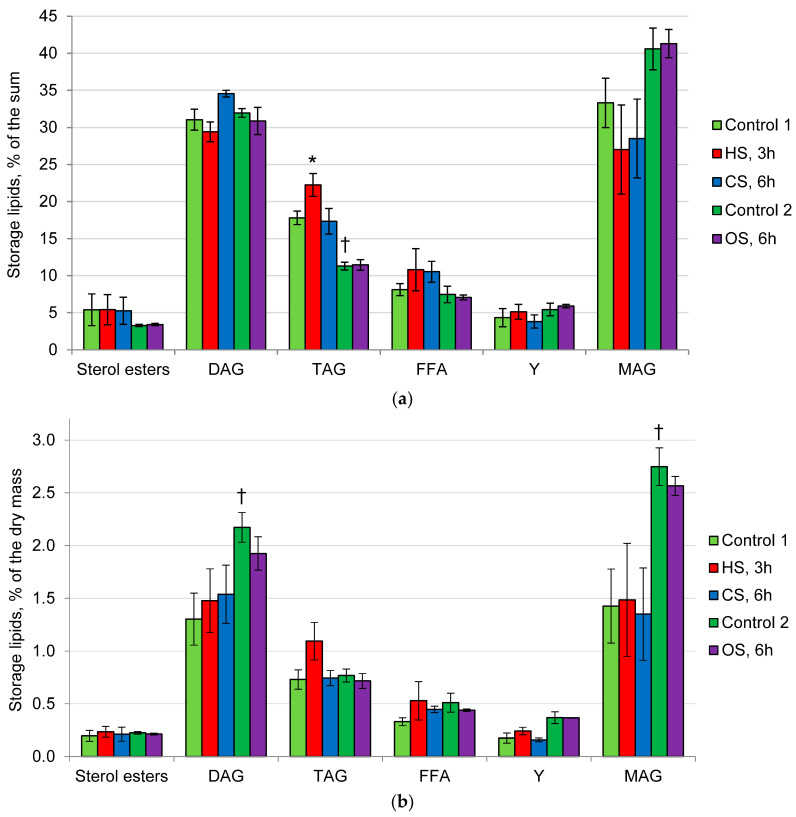
The profile of *P. gigantea* storage lipids: (**a**)—% of the dry mass; (**b**)—% of the sum. Means ± SEM are displayed, *n* = 3, SEM—standard error of the mean. Statistically significant difference (*p* ≤ 0.05) is indicated by (*)—between Control 1 and HS or CS; by (†) between Control 1 and Control 2. DAG—diacylglycerols, TAG—triacylglycerols, FFA—free fatty acids, Y—unidentified lipid, MAG—monoacylglycerols.

**Figure 4 ijms-25-03380-f004:**
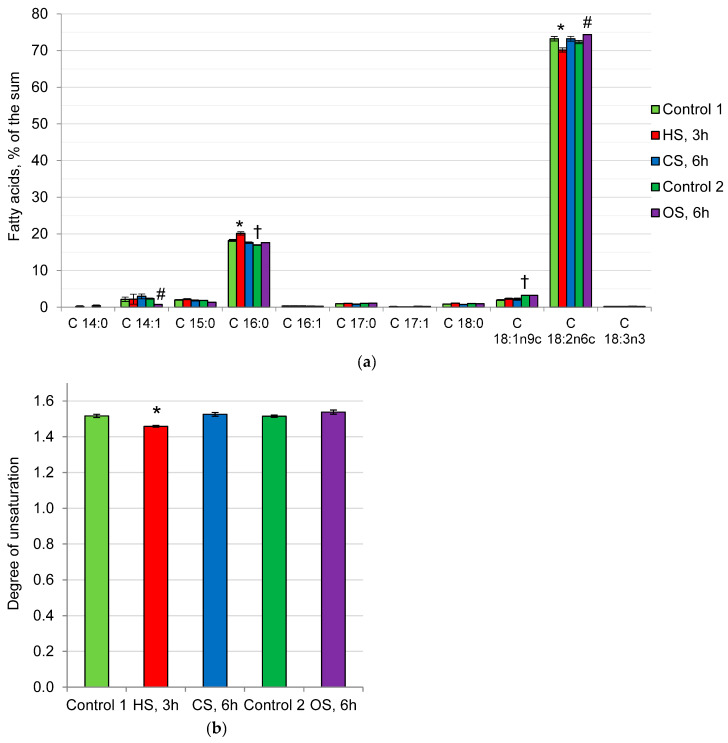
Fatty acids (**a**) and degree of unsaturation (**b**) of *P. gigantea* polar lipids. Means ± SEM are displayed, *n* = 3, SEM—standard error of the mean. Statistically significant difference (*p* ≤ 0.05) is indicated by (*)—between Control 1 and HS or CS; by (#) between Control 2 and OS; by (†) between Control 1 and Control 2.

**Table 1 ijms-25-03380-t001:** Amounts of membrane and storage lipids (% of dry weight) of *P. gigantea*.

	Control 1	HS, 3 h	CS, 6 h	Control 2	OS, 6 h
Membrane lipids, % of the dry mass	4.23 ± 0.18	3.66 ± 0.38	3.95 ± 0.29	6.08 ± 0.42	6.01 ± 0.44
Storage lipids, % of the dry mass	4.16 ± 0.69	5.06 ± 1.06	4.45 ± 0.79	6.79 ± 0.32 (†)	6.23 ± 0.21

Control 1 and Control 2—optimal conditions, HS—heat shock, CS—cold shock, OS—osmotic shock. Means ± SEM are displayed, *n* = 3, SEM—standard error of the mean. Statistically significant difference (*p* ≤ 0.05) is indicated by (†) between Control 1 and Control 2.

## Data Availability

Data are contained within the article.

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
