# Peer review of "Membrane Lipids and Osmolytes in the Response of the Acidophilic Basidiomycete Phlebiopsis gigantea to Heat, Cold, and Osmotic Shocks"

_ijms, 2024, doi:10.3390/ijms25063380_

Round 1

Reviewer 1 Report

Comments and Suggestions for Authors

This manuscript analyzed lipid & osmolyte contents in acidophilic fungi under heat, cold & osmotic shocks. Results suggested that trehalose level plays a key role during adaptation. The following issues should be addressed to improve the clarity of the manuscript.

1. Abstract lines 17-18; clear definition of %w/w and %total should be provided. e.g. weight of what? total CaP or total mass?

2. Exposure time to stressors is quite different, 6 h for cold & osmotic shocks but only 3 hr for heat shock. Why?

3. Figure legend & label are not clear (figures 1-4). What is the meaning of "% of the total"? Total mass or total CaP? Clear definition should be provided in the caption and also in the main text.

4. CaP contents in control 1 & control 2 were significantly different. An explanation or possible reason should be provided.

5. Figure 1; CaP content in control 2 and OS, 6h between figure (a) and (b) should go along with each other. Why is it significant (#) in (a) but not in (b)?

Reviewer 2 Report

Comments and Suggestions for Authors

This study is the report that the HS, CS, and OS gave the impact of the lipid composition of target cells. The variation of lipid composition against the external shock is the impotant information, from the viewpoint of lipidome. However, some of parts in the context was unclear. 

First, the reviewer did not clearly understand the motivation of the present study, although the data shown in this manuscript  was plainful. Why did the author compare the heat stress, cold stress, as well as osmosis stress? The authors mentioned into the stresses such as heat, cold, osmosis in terms of trehalose.

The increaseing of PA under control 2 and OS is interesting. Why was such an increase of PA under control 2 and OS condition induced?

In the conclusion, " the results obtained suggest that maintaining or increasing the level of trehalose can ensure the stability of the composition of membrane lipids under various stress impacts. (lines 279-281)" was stated. However, the present study did not show the case that the lipid composition was strongly altered in the case that the level of trehalose decleased. In such a sense, the experiment under adequate control systems was not performed. If the "control 2" is this, please add the explanation in the content.
